# Drug-related problems in hypertension and gestational diabetes mellitus: A hospital cohort

**Priscilla Karilline Vale Bezerra**[1⊙], **Jéssica Escorel Chaves Cavalcanti**[2‡], **Solimar Ribeiro Carlete Filho**[3‡], **Sarah Dantas Viana Medeiros**[4‡], **Antonio Gouveia Oliveira**[1,3⊙], **Rand Randall Martins**[1,2,3⊙] *

**1** Graduate Program of Pharmaceutical Science, Health Science Center, Federal University of Rio Grande do Norte, Natal, Rio Grande do Norte, Brazil, **2** Graduate Program in Sciences Applied to Women's Health, Maternidade Escola Januário Cicco, Health Sciences Center, Federal University of Rio Grande do Norte, Natal, Rio Grande do Norte, Brazil, **3** Pharmacy Department, Health Science Center, Federal University of Rio Grande do Norte, Natal, Rio Grande do Norte, Brazil, **4** School Maternity Januário Cicco, Health Science Center, Federal University of Rio Grande do Norte, Natal, Rio Grande do Norte, Brazil

⊙ These authors contributed equally to this work.
‡ These authors also contributed equally to this work.
* randrandall@gmail.com

**Data Availability Statement:** All relevant data are within the manuscript and its Supporting Information files.

## Abstract

### Objective

To characterize the drug-related problems (DRPs) in high-risk pregnant women with hypertension and gestational diabetes mellitus according to frequency, type, cause, and factors associated with their occurrence in the hospital setting.

### Methodology

This is an observational, longitudinal, prospective study that included 571 hospitalized pregnant women with hypertension and gestational diabetes mellitus using at least one medication. DRPs were classified according to the *Classification for Drug-Related Problems* (PCNE V9.00). In addition to descriptive statistics, a univariate and multivariate logistic regression model was employed to determine the factors associated with the DRPs.

### Results

A total of 873 DRPs were identified. The most frequent DRPs were related to therapeutic ineffectiveness (72.2%) and occurrence of adverse events (27.0%) and the main drugs involved were insulins and methyldopa. These were followed in the first five days of treatment by: the ineffectiveness of insulin (24.6%), associated with underdosage (12.9%) or insufficient frequency of administration (9.5%) and methyldopa associated with the occurrence of adverse reactions (40.2%) in the first 48h. Lower maternal age (OR 0.966, 95% CI 0.938–0.995, p = 0.022), lower gestational age (OR 0.966, 95% CI 0.938–0.996, p = 0.026), report of drug hypersensitivity (OR 2.295, 95% CI 1.220–4.317, p = 0.010), longer treatment

**Funding:** This study was funded by the National Counsel of Technological and Scientific Development (CNPq, Finance Code 001). The funders had no role in study design, data collection and analysis, decision to publish, or preparation of the manuscript.

**Competing interests:** The authors have declared that no competing interests exist.

time (OR 1.237, 95% CI: 1.147–1.333, p = 0.001) and number of prescribed medications (OR 1.211, 95% CI: 0.240–5.476, p = 0.001) were risk factors for occurrence of DRPs.

## Conclusion

DRPs are frequent in pregnant women with hypertension and gestational diabetes mellitus, and they are mainly related to therapeutic ineffectiveness and the occurrence of adverse events.

## Introduction

High-risk pregnancy is considered any clinical, obstetric, or social condition associated with pregnancy with a real or potential danger to the health and well-being of the maternal-fetal binomial [1]. Among the clinical conditions associated with high-risk pregnancies there is gestational hypertension (HG) and gestational diabetes mellitus (GDM), with an estimated prevalence between 1–20%, and high maternal-fetal morbidity and mortality [2].

The use of antihypertensives and insulin therapy are the mainstay of treatment for HG and GDM [3,4]. However, pregnancy brings about physiological changes in all systems impacting the pharmacokinetics and pharmacodynamics of many drugs [1]. Changes in oral absorption, increased volume of distribution, and changes in elimination profile are some examples that imply a dose-response relationship distinct from the non-pregnant adult [5]. These pregnant women are particularly vulnerable to adverse events such as occurrence of adverse drug reactions, thereby increasing uncertainties [6].

Few studies have investigated the nature and incidence of drug-related problems (DRPs) in hospitalized pregnant women. Recently, a study of 1117 Ethiopian pregnant women detected a 29% occurrence of one or more DRPs [7]. An Australian study of 241 hospitalized pregnant women detected a prevalence of DRPs (83%) identified over a short period of five weeks [8]. Another study investigated the occurrence of DRPs in pregnant women (42%) but listed medications used at home only [9]. A single Brazilian paper observed an occurrence of about 60% in DRPs in 600 puerperae with a DRP diagnosis of preeclampsia [10].

However, despite HG and GDM being the main causes of high-risk pregnancies, as far as we know, there are no studies that characterize DRPs in these patients. Given the scarcity of information, we conducted a prospective study with the objective of characterizing DRPs in pregnant women with HG and GDM according to frequency, type, cause, and factors associated with their occurrence in the hospital setting.

## Methodology

### Study design

An observational prospective cohort study was conducted in a maternity school in the municipality of Natal/Brazil, the institution has 22 beds for high-risk pregnancies and about 1,919 annual admissions. We included consecutively, between September 2019 to July 2022, pregnant women of all ages with a diagnosis of GH and/or GDM with hospitalization time longer than 24 hours and with a prescription for one or more medications. Readmitted deaf pregnant women and those hospitalized for diagnostic procedures only were excluded. The study was approved by the Institutional Review Board of the University Hospital Onofre Lopes, according to the determinations of Resolution CNS No. 466/12 of the National Health Council, with

ruling No. 3,483.151/2019 and written informed consent was obtained from the all patients. In the case of underage, written informed consent was obtained from the guardians too.

## Data collection

Data were collected daily through interviews with pregnant women and consultation of medical records. Pregnant women were recruited consecutively upon admission, and the number of patients assessed simultaneously was limited to a maximum of 12. This limit was defined in a pilot study as it was adequate for the capacity of the research team and allowed higher quality data to be obtained during collection. Clinical and demographic variables such as admission diagnosis, gestational age in weeks (GA), comorbidity, allergy, age in years, number of previous pregnancies, history of miscarriage, and laboratory variables (hemogram, C-reactive protein, electrolytes, total protein and fractions, as well as glycemic, hepatic and renal profile) were collected.

Admission diagnoses were grouped according to the International Classification of Diseases version 10 (ICD—10) [11]. The prescribed medications were classified according to the *Anatomical Therapeutic Chemical Code Classification System* (ATCC) [12], as well as the appropriateness of the dose and frequency of administration.

In this study, DRPs are defined as "event or circumstance involving drug therapy that actually or potentially interferes with the desired health outcome" [13]. The DRPs were categorized by type of problem and cause according to those proposed by the *Classification for Drug-Related Problems* (PCNE), version 9.00 [13]. The problems were grouped into three main domains: treatment efficacy, when the drug was not or may not have had the expected effect; adverse reaction, when the patient experienced or was at high risk of an adverse drug event; and the cost of treatment, when other cost-effective drugs were available for treatment. DRPs that did not fall into any of these three categories were classified as problems with no defined category. The cause categories were grouped into nine main domains: drug selection, drug form, dose selection, treatment duration, dispensing, drug use process, patient-related, patient-transfer-related, and other. It is important to highlight that a problem can be caused by more than one cause, especially in the hospital environment.

The detection of the DRPs was carried out through an active search following four steps: prescription analysis, active search in medical records, pharmaceutical anamnesis, and classification of the DRPs.

- Prescription analysis (Step 1)—the indication and dosage was evaluated for prescription errors (dose, frequency and indication) and potential drug interactions.

- Active search in medical records (Step 2)—after prescription analysis, the medical records were investigated for clinical and laboratory parameters related to ineffectiveness and unreliability of medications. Additionally, modifications, substitutions, and/or suspension of medications were also warning signs.

- Daily interviews (Step 3)—after the start of therapy, all patients were questioned daily regarding the occurrence of symptoms that could correspond to adverse reactions and treatment failure. The questions were directed according to the previous evaluation of the day's prescription and medical records.

- Classification of DRPs (Step 4)—The suspected identification of DRPs that occurred during the hospitalization was classified according to the PCNE.

Data collection was performed by two previously trained pharmacists (PKVB and JECC) with the help of pharmacy students. Every day the team performed steps 1, 2, 3 and 4 of the

study, however, the classification of DRPs (Step 4) was done only by the researchers PKVB and JECC (see S1 Table). In case of disagreement in the classification of DRPs and to reduce the occurrence of rater bias, a third clinical pharmacist (RRM) was consulted. A pilot study with 10 patients was run to check the acceptability and consistency of the data collection instrument two weeks before the actual data collection.

## Statistical analysis

The sample size was set at 600 individuals which ensures a maximum error of the estimates of ± 4 percentage points with 95% confidence. We employed the following formula to calculate the sample size:

$$Sample\ size = \frac{Z_{1-\alpha/2}{}^2 p(1-p)}{d^2}$$

Here:

$Z_{1-\alpha/2}$ = 1.96 (standard normal variable considering a type 1 error of 5% and p <0.05).

p = Expected proportion of DRP in the population. As there are no similar studies available, we opted for a proportion that enables the largest sample size (50%).

d = Absolute error (4%).

The data analysis of the quantitative stage of this research was conducted using descriptive and inferential statistics with Stata version 15 software (Stata Corporation, College Station, TX, USA). The descriptive statistics included the median and 25th and 75th percentiles (p25-75%); absolute frequency and percent proportion; and mean and standard deviation according to the type of variable under analysis.

The incidence of DRPs was expressed as incidence density (number of DRPs per 1,000 patient-days) and 95% confidence interval (CI). The distribution of the occurrence of DRPs and respective ATC classes were presented as an incidence rate (DRPs per 100 patients) for the first five days of hospitalization. To determine the risk factors for the occurrence of DRPs, the association of the occurrence of one or more DRPs with each clinical variable of the individual patients was analyzed by univariate logistic regression, estimating the respective odd-ratios (OR) and 95% confidence intervals. Variables that exhibited a significant association with a p-value < 0.10 were included in a multivariate logistic regression model. The stepwise backward variable selection method, with a significance level of p < 0.05, was used to identify independent factors associated with the occurrence of DRPs in pregnant women.

## Results

During the study period, approximately 5,607 high-risk pregnant women were admitted to the institution between September 2019 and July 2022. However, due to the SARS-COV 2 pandemic, data collection was temporarily suspended from March 2020 to April 2021 (2,152 pregnant women attended). In the non-pandemic period, the institution admitted 3,455 pregnant women, of these, 30 refused to participate in research and 2,854 exceeded the capacity of the research team. We selected 571 patients with a median age of 31 years (p25–75% = 26 to 36 years) and a gestational age of 34 weeks (p25–75% = 29 to 36 weeks). Hypertensive syndromes were the leading causes of admission diagnosis (398; 69.7%), followed by GDM (326; 57.1%). Among the patients admitted, 305 (53.5%) had one to two previous deliveries and 169 (29.6%) had a history of miscarriage. The mean treatment time was 6.0 ± 5.1 and the mean total number of drugs prescribed per patient was 7.7 ± 2.7. The incidence rate of DRPs per 1000 patient-days was 50.4 (95% confidence interval (95% CI) 42.4–60.1) with a predominance in the first

**Table 1. Characterization of the study population (n = 571).**

| Features | Values | |
|---|---|---|
| Age in years (med, p25–75%) | 31 | 26–36 |
| Gestational age in weeks (med, p25–75%) | 34 | 29–36 |
| Reported drug hypersensitivity (n, %) | 83 | 14.5 |
| Admission diagnosis (n, %) | | |
| Hypertensive syndromes | 398 | 69.7 |
| Gestational diabetes mellitus | 326 | 57.1 |
| Urinary tract infections and vaginosis | 40 | 7.0 |
| Fetal and placental abnormalities | 32 | 5.6 |
| Parity (n, %) | | |
| 1st pregnancy | 172 | 30.2 |
| 1 to 2 previous births | 305 | 53.5 |
| 3 or more previous births | 93 | 16.3 |
| History of previous miscarriage (n, %) | 169 | 29.6 |
| Treatment time in days (m, sd) | 6.0 | 5.1 |
| Medications per patient (m, sd) | 7.7 | 2.7 |
| Incidence rate (DRPs per 1000 patient-days, 95% CI) | 50.4 | 42.4–60.1 |
| DRP Prevalence (n, %) | 364 | 63.8 |
| Period of the occurrence of DRPs (n, %) | | |
| First 48h | 594 | 68.0 |
| Between the 3rd and 5th day of hospitalization | 176 | 20.2 |
| 6th day or more from admission | 103 | 11.8 |
| Days of DRPs duration (med, p25–75%) | 4.0 | 2–7 |

Median (med), 25th and 75th percentiles (p25–75%), absolute and relative frequency (n, %), confidence interval (95%CI), mean (m) and standard deviation (sd).

48h (594, 68.0%), and with a median DRP duration of 4 days (p25: 2, p75: 7). We detected a prevalence of 364 (63.8%) DRPs (Table 1).

Table 2 describes the profile of DRPs in regard to type, a total of 873 occurrences were identified. Most of the DRPs were related to therapeutic ineffectiveness (P1: 630, 72.2%), especially non-optimal treatment (P1.2: 541; 62.0%). Also noteworthy was the occurrence of adverse events (P2.1: 236, 27.0%).

In relation to the causes of DRPs and the **main subgroups involved** (Table 3), we highlight the role of insulins (A10A) and centrally acting anti-adrenergics (C02A), primarily methyldopa. Insulin-related ineffectiveness DRPs (P1: 214, 24.6%) had as main causes inadequate dose selection, mainly associated with underdosage (C3.1: 112, 12.9%) and insufficient administration frequency (C3.3: 83, 9.5%). Methyldopa was frequently related to the occurrence of adverse reactions (350, 40.2%). When the incidence rate in the first five days of treatment was evaluated (Fig 1), there was a higher occurrence in the first 48 hours, with a significant decrease after the 4th day of hospitalization. Causes of DRPs on the first day were: underdosage of insulin (6.8 DRPs per 100 patients, CI95% 4.9–9.2), frequency of insufficient insulin administration (5.3 per 100 patients, CI95% 3.6–7.4) and adverse reactions associated with methyldopa (8.2 per 100 patients, CI95% 6.1–10.9).

Univariate analysis (Table 4) identified lower gestational age (OR 0.948, CI95% 0.921–0.967, p = 0.001), report of drug hypersensitivity (OR 2.692, CI95% 1.515–4.785, p = 0.001), urinary tract infections and vaginosis (OR 2.849, CI95% 1.234–6.560, p = 0.014), fetal and placental abnormalities (OR 0.420, CI95% 0.204–0.863, p = 0.018), longer hospital stay (OR

**Table 2. Profile of identified Drug-Related Problems (DRPs) according to the Pharmaceutical Care Network Europe (PCNE) V9.00 classification.**

| Primary domain | Code | Detailed Classification | N | % |
|---|---|---|---|---|
| Treatment effectiveness There is a (potential) problem with the (lack of) effect of the pharmacotherapy | | P1 | | |
| | P1.1 | No effect of drug treatment despite correct use | 85 | 9.7 |
| | P1.2 | Effect of drug treatment not optimal | 541 | 62.0 |
| | P1.3 | Untreated symptoms or indication | 4 | 0.5 |
| Treatment safety Patient suffers, or could suffer, from an adverse drug event | | P2 | | |
| | P2.1 | Adverse drug event (possibly) occurring | 236 | 27.0 |
| Other | | P3 | | |
| | P3.1 | Problem with cost-effectiveness of the treatment | 0 | 0.0 |
| | P3.2 | Unnecessary drug-treatment | 0 | 0.0 |
| | P3.3 | Unclear problem/complaint | 7 | 0.8 |
| Total | | | 873 | 100.0 |

PCNE, Pharmaceutical Care Network Europe.

1.328, CI95% 1.236–1.428, p = 0.001), and higher number of prescribed medications (OR 1.403, CI95% 1.292–1.523, p = 0.001) associated with the occurrence of DRPs. However, after multivariate analysis, lower maternal age (OR 0.966, 95% CI 0.938–0.995, p = 0.022), lower

**Table 3. Causes of Drug-Related Problems (DRPs) (n = 873) according to the Pharmaceutical Care Network Europe (PCNE) V9.00 classification and main sub-groups involved.**

| Primary Domain | Code | Detailed Classification | N | % | Major ATC | N | % |
|---|---|---|---|---|---|---|---|
| Drug selection | C1.2 | Inappropriate drug | 6 | 0.7 | C02A | 4 | 0.5 |
| | C1.3 | No indication for drug | 1 | 0.1 | J01C | 1 | 0.1 |
| | C1.6 | No or incomplete drug treatment in spite of existing indication | 19 | 2.2 | A10A | 7 | 0.8 |
| Drug form | C2.1 | Inappropriate drug form (for this patient) | 4 | 0.5 | C03A | 2 | 0.2 |
| Dose selection | C3.1 | Drug dose too low | 112 | 12.9 | A10A | 70 | 8.0 |
| | C3.2 | Drug dose too high | 19 | 2.2 | A10A | 13 | 1.5 |
| | C3.3 | Dosage regimen not frequent enough | 83 | 9.5 | A10A | 52 | 6.0 |
| | C3.4 | Dosage regimen too frequent | 8 | 0.9 | C02A | 3 | 0.3 |
| | C3.5 | Dose timing instructions wrong, unclear or missing | 226 | 25.9 | B03A | 144 | 16.5 |
| Dispensing | C5.2 | Necessary information not provided | 3 | 0.3 | A04A | 1 | 0.1 |
| Drug use process | C6.4 | Drug not administered at all | 1 | 0.1 | G01A | 1 | 0.1 |
| | C6.5 | Wrong drug administered | 1 | 0.1 | V03A | 1 | 0.1 |
| Other | C9.1 | No or inappropriate outcome monitoring (incl. TDM) | 4 | 0.5 | A10A | 4 | 0.5 |
| | C9.2 | Other cause; specify | 350 | 40.2 | C02A | 66 | 7.6 |
| | C9.3 | No obvious cause | 34 | 3.9 | A10A | 6 | 0.7 |

ATC, Anatomical Therapeutic Chemical Classification System; PCNE, Pharmaceutical Care Network Europe. C02A - centrally acting antiadrenergics, J01C - anti-infectives for systemic use, A04A - antiemetics and antinausea, A10A - insulins and analogues, C03A - thiazide diuretics, B03A - iron preparations, G01A - combinations with corticosteroids, V03A - other classes.

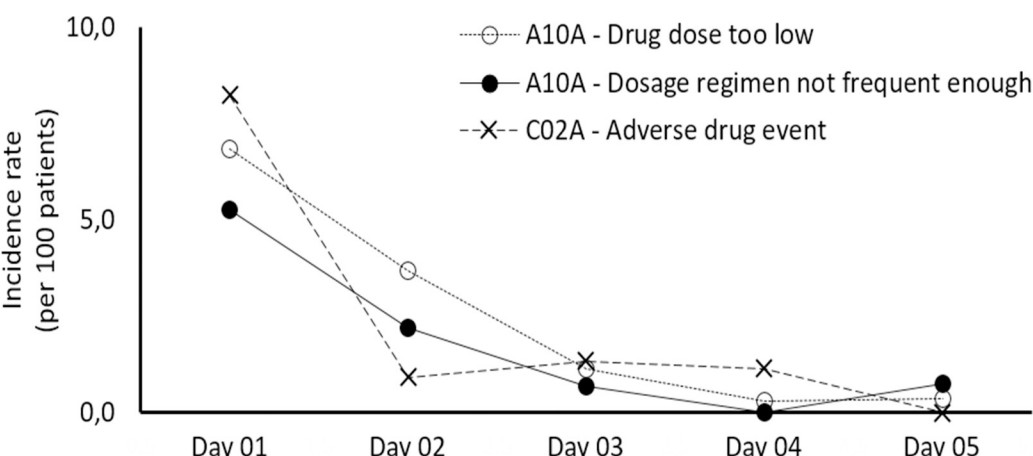

**Fig 1. Distribution of the incidence of causes of Drug-Related Problems (DRPs) associated with the prescription of insulins (A10A) and centrally acting antihypertensives (C02A) in the first five days of hospitalization.** Legend: A10A - insulins and analogues, C02A - centrally acting antiadrenergic agents.

gestational age (OR 0.966, 95% CI 0.938–0.996, p = 0.026), report of drug hypersensitivity (OR 2.295, 95% CI 1.220–4.317, p = 0.010), longer treatment time (OR 1.237, 95% CI: 1.147–1.333, p = 0.001) and number of prescribed medications (OR 1.211, 95% CI: 0.240–5.476, p = 0.001) were shown to be risk factors for occurrence of DRPs.

## Discussion

In this prospective cohort study based on a representative sample of hypertensive and diabetic pregnant women, the high occurrence of DRPs primarily related to therapeutic ineffectiveness and adverse reactions is highlighted. The detection of DRPs predominated in the first 48 hours, highlighting ineffectiveness associated with insufficient insulin dose and adverse reactions to methyldopa. Lower maternal age, lower gestational age, previous history of drug hypersensitivity, longer treatment time, and number of prescribed drugs were associated with the occurrence of DRPs.

**Table 4. Risk factors associated with the occurrence of Drug-Related Problems (DRPs) in hospitalized pregnant women.**

| Features | Univariate analysis | | | Mulitvariate analysis | | |
|---|---|---|---|---|---|---|
| | OR | 95%CI | | P | OR | 95%CI | | P |
| Age in years | 0.979 | 0.954 | 1.004 | 0.105 | 0.966 | 0.938 | 0.995 | 0.022 |
| Gestational age in weeks | 0.948 | 0.921 | 0.976 | <0.001 | 0.966 | 0.938 | 0.996 | 0.026 |
| Report of drug hypersensitivity | 2.692 | 1.515 | 4.785 | 0.001 | 2.295 | 1.220 | 4.317 | 0.010 |
| Hypertensive syndromes | 0.940 | 0.648 | 1.365 | 0.745 | - | - | - | - |
| Gestational diabetes mellitus | 1.037 | 0.735 | 1.454 | 0.835 | - | - | - | - |
| Urinary Tract Infections and Vaginosis | 2.849 | 1.237 | 6.560 | 0.014 | - | - | - | - |
| Fetal and placental abnormalities | 0.420 | 0.204 | 0.863 | 0.018 | - | - | - | - |
| Parity | 1.003 | 0.886 | 1.135 | 0.960 | - | - | - | - |
| History of previous miscarriage | 1.100 | 0.695 | 1.468 | 0.960 | - | - | - | - |
| Treatment time in days | 1.328 | 1.236 | 1.428 | <0.001 | 1.237 | 1.147 | 1.333 | <0.001 |
| Number of prescribed drugs | 1.403 | 1.292 | 1.523 | <0.001 | 1.211 | 0.240 | 5.476 | <0.001 |

OR, Odds-ratio; CI, confidence interval.

Several classification systems have been created by different research groups for the identification of DRPs, but what they all have in common is the identification of problems associated with drug selection, dose selection, adverse events and therapeutic adherence [14]. The use of these classification systems makes it possible to compare different scenarios regarding the occurrence of DRPs and to elaborate conduct protocols, which is especially supportive for pharmaceutical care. Among these systems, the PCNE classification system presents a greater detailing of the causes of DRPs and is frequently involved in drug review studies [15]. Regarding the characterization of DRPs in hospitalized pregnant women, the literature is quite scarce.

With a similar approach and findings to ours, we highlight an Australian cohort with 241 hospitalized pregnant women evaluated daily by a clinical pharmacy service [8]. Using the classification system proposed by Cipolle, Strand & Morley, the authors identified the occurrence of DRPs in about 80% of patients with a predominance of posological inadequacies as the main causes. In contrast, a study of 1117 Ethiopian pregnant women detected an occurrence of 29% of one or more DRPs while also using Cipolle, Strand & Morley's classification [7]. The main detected DRP was the need for additional therapy (73%). However, this differs from our sample in that the women were much younger (median age 25 years) and almost half lived in rural areas. In two Norwegian maternity hospitals, Smedberg et al. [9], investigated the occurrence of DRPs (PCNE classification) in pregnant women (at least one DRP in 42% of pregnant women), but reported on the drugs used at home only. A Brazilian cross-sectional study conducted in two maternity hospitals with 600 women, detected the main DRPs (PCNE classification) were related to medication not being administered, untreated problems and therapeutic ineffectiveness, however, only puerperal women were evaluated [10].

These authors mainly evaluated pregnant women at the time of delivery and postpartum, with spontaneous abortions (miscarriages), hyperemesis gravidarum, allergies, respiratory tract infections and preeclampsia predominating as main diagnoses. To the best of our knowledge, our study is the first to address pregnant women hospitalized due to complications of HG and GDM. Therefore, the higher occurrence of observed DRPs is probably due to the greater severity of these pregnant women at admission. GDM is characterized by maternal hyperglycemia affecting the structure and vascularization of the placenta, and is associated with maternal-fetal complications and increased risk of perinatal morbidity and mortality [16]. Pregnant women with GDM are more likely to be hospitalized [17] and require insulin therapy for strict glycemic control [18]. Additionally, women with GDM are at increased risk of gestational hypertension [19]. Hypertensive crises can decrease placental perfusion, resulting in fetal growth restriction, preeclampsia or eclampsia [20]. This aggravation of GH, leads to episodes of emergency care and hospitalization with 7.2% of total hospitalizations in pregnancy [21].

Therapeutic inefficacy associated with insufficient doses and the occurrence of adverse drug reactions (ADRs) compose the identified profile of DRPs, highlighting that seven out of ten DRPs occur in the first 48 hours. One hypothesis for the high occurrence of DRPs on admission would be the greater severity of the clinical picture. Generally, these pregnant women are hospitalized with marked hyperglycemia and/or high blood pressure values that require immediate control.

Insulin was the main drug related to DRPs of effectiveness, in which underdosage and insufficient frequency of administration stand out. Glycemic control through insulin administration is the treatment of first choice. The Brazilian guidelines, adopted by the evaluated institution, recommend the initial dose of NPH insulin 0.3 IU/kg divided into two administrations [22], but other authors recommend doses between 0.7–2.0 IU/kg for initial glycemic control [23–25]. Although initial insulin doses are commonly selected based on weight, Nadeau et al. [25] suggest that baseline blood glucose may be a more relevant parameter since placental hormones induce a more resilient picture.

Methyldopa was the drug most implicated in the occurrence of ADR, and has been frequently observed in hospitalized high-risk pregnant women [26]. It is a priority to reduce blood pressure in hospitalized pregnant women, and this requires optimized doses, methyldopa presents a wide therapeutic margin that makes it possible to double the daily dose in 48 hours [27]. However, its action at the central nervous system level can cause ADRs such as sedation and headache [26,28]. Of the ADRs detected, none presented significant risk to the pregnant woman and no intervention was necessary.

From the point of view of the detection of DRPs within the first five days of treatment, the first 48 hours are critical due to hyperglycemia and hypertension. These situations require optimized doses of insulin and antihypertensives, primarily methyldopa. However, unlike methyldopa, where higher doses used on day 1 cause low-severity ADRs that tend to decrease on day 2, probably due to masking of symptoms with the administration of other medications, insulin tends to be employed at lower doses due to its higher risk of causing adverse outcomes. However, this implies a delay in achieving glycemic control, which is usually achieved by day 3 of hospitalization.

In multivariate analysis, some characteristics at the time of admission were associated with the occurrence of DRPs, specifically younger age and shorter gestation time. The occurrence of DRPs related to younger age is a controversial finding; pregnancies at older ages imply a higher risk of complications and would be a risk factor for the occurrence of DRPs [8,29,30]. Only one author has associated the occurrence of adverse reactions with younger pregnant women when hospitalized [31]. On the other hand, the higher occurrence of DRPs in earlier stages of pregnancy is potentially associated with greater clinical severity, since the need for hospitalization at the beginning of the third trimester indicates more severe conditions that are difficult to manage pharmacologically [1].

Additionally, longer hospitalization, greater number of prescribed medications and history of drug allergy presented an increased risk for the occurrence of DRPs. Other authors have also identified longer hospital stays and greater use of medications as risk factors for the occurrence of DRPs in pregnant and postpartum women [8,10]. This is justified by the higher exposure to medications which is also observed in general hospitals [32]. A report of drug allergy before pregnancy is associated with higher occurrence of prematurity and fetal growth restriction [33,34], theoretically, pregnant women would be more prone to greater use of medication and, consequently, higher occurrence of DRPs.

The main limitation of our study was that the collection was performed in a single institution. However, some methodological characteristics validate the results, such as the use of the PCNE classification system, which is frequently applied in studies involving the identification and categorization of DRPs and is a tool frequently used in hospital practice [15], the large sample size, the prospective cohort design, and the detection of DRPs by daily active search.

The incidence, characterization and factors associated with the occurrence of DRPs allows the multidisciplinary team to provide information for the planning of actions aimed at the safe use of these medications, expanding their repertoire and introducing new practices for the rational use of medicines during high-risk pregnancy. Future research to characterize the best approach for glycemic control of these pregnant women is necessary.

## Conclusion

In summary, we observed that DRPs are frequent in pregnant women with gestational hypertension and gestational diabetes mellitus, and this can lead to therapeutic ineffectiveness and adverse events. The drugs most involved with DRPs are insulins and methyldopa. Inadequate insulin dose selection and methyldopa adverse reactions are the main causes of DRPs in the

first 48 h of treatment, with a tendency to decrease by day 5. Finally, younger maternal age, lower gestational age, report of drug hypersensitivity, length of treatment, and the number of drugs prescribed are risk factors associated with the occurrence of DRPs.

## Supporting information

**S1 Table. Parameters used for classification of effectiveness and safety in Drug Related Problems (DRP).**
(DOCX)

## Acknowledgments

The authors thank the Teaching and Research Management of the Januário Cicco Maternity School for making this research possible and the Pharmacy students Arlan Florencio, Conceição Lira, Marilia Martins, Mike Costa, Solimar Filho, Anny Silva, Gabriela Oliveira, and Luiz Mendes for collecting the data.

## Author Contributions

**Conceptualization:** Antonio Gouveia Oliveira, Rand Randall Martins.

**Data curation:** Rand Randall Martins.

**Formal analysis:** Antonio Gouveia Oliveira, Rand Randall Martins.

**Funding acquisition:** Rand Randall Martins.

**Investigation:** Priscilla Karilline Vale Bezerra, Jéssica Escorel Chaves Cavalcanti, Solimar Ribeiro Carlete Filho, Sarah Dantas Viana Medeiros, Rand Randall Martins.

**Methodology:** Priscilla Karilline Vale Bezerra, Jéssica Escorel Chaves Cavalcanti, Solimar Ribeiro Carlete Filho, Sarah Dantas Viana Medeiros, Antonio Gouveia Oliveira, Rand Randall Martins.

**Project administration:** Priscilla Karilline Vale Bezerra, Jéssica Escorel Chaves Cavalcanti, Sarah Dantas Viana Medeiros, Antonio Gouveia Oliveira, Rand Randall Martins.

**Resources:** Rand Randall Martins.

**Software:** Rand Randall Martins.

**Supervision:** Priscilla Karilline Vale Bezerra, Jéssica Escorel Chaves Cavalcanti, Solimar Ribeiro Carlete Filho, Sarah Dantas Viana Medeiros, Rand Randall Martins.

**Validation:** Rand Randall Martins.

**Visualization:** Rand Randall Martins.

**Writing – original draft:** Priscilla Karilline Vale Bezerra, Antonio Gouveia Oliveira, Rand Randall Martins.

**Writing – review & editing:** Priscilla Karilline Vale Bezerra, Jéssica Escorel Chaves Cavalcanti, Solimar Ribeiro Carlete Filho, Sarah Dantas Viana Medeiros, Antonio Gouveia Oliveira, Rand Randall Martins.

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
