## [Decision Letter · Decision Letter 0]

25 Jan 2023

PONE-D-22-28909Drug-related problems in hypertension and gestational diabetes mellitus: a hospital cohortPLOS ONE

Dear Dr. Martins,

Thank you for submitting your manuscript to PLOS ONE. After careful consideration, we feel that it has merit but does not fully meet PLOS ONE’s publication criteria as it currently stands. Therefore, we invite you to submit a revised version of the manuscript that addresses the points raised during the review process.

We look forward to receiving your revised manuscript.

Kind regards,

Nnabuike Chibuoke Ngene, Dip HIV Med; MMed(FamMed); FCOG; MMed(O&G); Ph.D

Academic Editor

PLOS ONE

and https://journals.plos.org/plosone/s/file?id=ba62/PLOSOne_formatting_sample_title_authors_affiliations.pdf.

“This study was financed in part by the Coordenação de Aperfeiçoamento de Pessoal de Nível Superior – Brasil (CAPES) - Finance Code 001”

3. Thank you for stating the following in the Acknowledgments/ Funding Section of your manuscript:

“This study was financed by the Coordenação de Aperfeiçoamento de Pessoal de Nível 25 Superior – Brasil (CAPES) - Finance Code 001.”

“This study was financed in part by the Coordenação de Aperfeiçoamento de Pessoal de Nível Superior – Brasil (CAPES) - Finance Code 001”

Additional Editor Comments:

Statistical analysis, sentence: “The sample size was set at 600 individuals, which ensures a maximum error of the estimates of ± 4 percentage points with 95% confidence.” Was the sample size determined a priori or post hoc? Provide details about how 4% was calculated (including the software/formula used).

In statistical analysis, the authors mentioned absolute and relative frequency. Provide additional explanation. Did the authors report relative frequency in the manuscript?

In statistical analysis, state the method used for selecting the variables that were included in the multivariable regression analysis.

Reviewers' comments:

Reviewer's Responses to Questions

**Comments to the Author**

1. Is the manuscript technically sound, and do the data support the conclusions?

Reviewer #1: Yes

Reviewer #2: Yes

2. Has the statistical analysis been performed appropriately and rigorously? 

Reviewer #1: Yes

Reviewer #2: Yes

3. Have the authors made all data underlying the findings in their manuscript fully available?

Reviewer #1: Yes

Reviewer #2: Yes

4. Is the manuscript presented in an intelligible fashion and written in standard English?

Reviewer #1: Yes

Reviewer #2: Yes

5. Review Comments to the Author

Reviewer #1: Thank you for asking me to review this article. The article is not only interesting but quite informative as it appears to bring to our consciousness some of the obscure factors affecting drug effectiveness in high risk pregnancy. The methodology is appropriate for the research objectives. The use of a single institution for data collection seems to be a major concern, but this can be addressed with a multi centre study in future. I would recommend this article for publication

Reviewer #2: In the abstract, the following, the univariate analysis is not mentioned. Also Insulin and methyldopa are not pharmacological classes. Please check and correct

52 addition to descriptive statistics, a multivariate logistic regression model was employed to

53 determine the factors associated with the DRPs. Results: A total of 873 DRPs were identified.

54 The most frequent DRPs were related to therapeutic ineffectiveness (72.2%) and occurrence of

55 adverse events (27.0%) and the main pharmacological classes involved were insulins and

56 methyldopa.

6. PLOS authors have the option to publish the peer review history of their article (what does this mean?). If published, this will include your full peer review and any attached files.

Reviewer #1: **Yes: **Ekwuazi Kingsley Emeka

Reviewer #2: **Yes: **Modupe Olufunmilayo Ogunrombi

---

## [Author Response · Author response to Decision Letter 0]

24 Feb 2023

23th February 2023

Dear Dra Irles,

In this letter, we detail the changes introduced in the manuscript “Drug-related problems in hypertension and gestational diabetes mellitus: a hospital cohort” in response to the reviewers’ comments. All reviewers’ comments were accepted and we appreciate the important collaborations. 

Yours sincerely, 

Rand Randall Martins, PharmD. PhD.

Corresponding Author

Journal Requirements 

Response: The style template was readequate as suggested.

2. Thank you for stating the following financial disclosure: “This study was financed in part by the Coordenação de Aperfeiçoamento de Pessoal de Nível Superior – Brasil (CAPES) - Finance Code 001”. Please state what role the funders took in the study. If the funders had no role, please state: ""The funders had no role in study design, data collection and analysis, decision to publish, or preparation of the manuscript.""

Response: The cover letter was readjusted as suggested.

3. Thank you for stating the following in the Acknowledgments/ Funding Section of your manuscript: “This study was financed by the Coordenação de Aperfeiçoamento de Pessoal de Nível 25 Superior – Brasil (CAPES) - Finance Code 001.”

“This study was financed in part by the Coordenação de Aperfeiçoamento de Pessoal de Nível Superior – Brasil (CAPES) - Finance Code 001”

Response: The manuscript has been corrected as per the suggestions. 

Response: The references were thoroughly reviewed and corrected to align with the formatting guidelines of the journal. In order to meet the publication criteria, minor changes were made to the formatting, including replacing one reference with a more current and relevant source. (page 25, line 478-486).

Deleted reference:

Ghosh KR, Akhter S, Das AK, Naher N, Paul SR, Islam B. Outcome of Labetalol and Methyldopa as Oral Antihypertensive Agent in the Treatment of Pregnancy Induced Hypertension. Mediscope. 2021;8(1):19-26. doi: https://doi.org/10.339/mediscope.v8i1.52200

Added reference:

Van de Vusse D, Mian P, Schoenmakers S, Flint RB, Visser W, Allegaert K, et al. Pharmacokinetics of the most commonly used antihypertensive drugs throughout pregnancy methyldopa, labetalol, and nifedipine: a systematic review. Eur J Clin Pharmacol. 2022;78(11):1763-1776. doi: 10.1007/s00228-022-03382-3 

5. Statistical analysis, sentence: “The sample size was set at 600 individuals, which ensures a maximum error of the estimates of ± 4 percentage points with 95% confidence.” Was the sample size determined a priori or post hoc? Provide details about how 4% was calculated (including the software/formula used).

Response: The text was changed to (page 8, line 163-169):

“The sample size was set at 600 individuals, which ensures a maximum error of the estimates of ± 4 percentage points with 95% confidence. We employed the following formula to calculate the sample size: 

Sample size=(〖Z_(1-∝/2)〗^2 p(1-p))/d^2 

Here:

Z 1- α/2 = 1.96 (standard normal variable considering a type 1 error of 5% and p <0.05).

p = Expected proportion of DRP in the population. As there are no similar studies available, we opted for a proportion that enables the largest sample size (50%).

d = Absolute error (4%).”

6. In statistical analysis, the authors mentioned absolute and relative frequency. Provide additional explanation. Did the authors report relative frequency in the manuscript?

Response: Thanks for the observation, the term used was imprecise. The text was changed to (page 8, line 173):

“The descriptive statistics included the median and 25th and 75th percentiles (p25-75%); relative and absolute frequency and percent proportion; and mean and standard deviation according to the type of variable under analysis.”

7. In statistical analysis, state the method used for selecting the variables that were included in the multivariable regression analysis.

Response: The text was changed to (page 9, line 182-188):

“All variables with an association test with p-value <0.10 were included in a multivariate logistic regression model, and a significance level of p<0.05 was then adopted to identify the factors independently associated with the occurrence of DRPs in pregnant women. Variables that exhibited a significant association with a p-value < 0.10 were included in a multivariate logistic regression model. The stepwise backward variable selection method, with a significance level of p < 0.05, was used to identify independent factors associated with the occurrence of DRPs in pregnant women.”

Reviewer #1

8. Thank you for asking me to review this article. The article is not only interesting but quite informative as it appears to bring to our consciousness some of the obscure factors affecting drug effectiveness in high risk pregnancy. The methodology is appropriate for the research objectives. The use of a single institution for data collection seems to be a major concern, but this can be addressed with a multi centre study in future. I would recommend this article for publication.

Reviewer #2

9. In the abstract, the following, the univariate analysis is not mentioned. Also Insulin and methyldopa are not pharmacological classes. Please check and correct.

Response: Following the reviewer's suggestion, the text was changed to (page 3, line 51-56):

“In addition to descriptive statistics, a univariate and multivariate logistic regression model was employed to determine the factors associated with the DRPs. Results: A total of 873 DRPs were identified. The most frequent DRPs were related to therapeutic ineffectiveness (72.2%) and occurrence of adverse events (27.0%) and the main drugs and the main pharmacological classes involved were insulins and methyldopa.”

---

## [Decision Letter · Decision Letter 1]

22 Mar 2023

Drug-related problems in hypertension and gestational diabetes mellitus: a hospital cohort

PONE-D-22-28909R1

Dear Dr. Martins,

We’re pleased to inform you that your manuscript has been judged scientifically suitable for publication and will be formally accepted for publication once it meets all outstanding technical requirements.

Kind regards,

Nnabuike Chibuoke Ngene, Dip HIV Med; MMed(FamMed); FCOG; MMed(O&G); Ph.D

Academic Editor

PLOS ONE

Additional Editor Comments (optional):

Reviewers' comments:

Reviewer's Responses to Questions

**Comments to the Author**

1. If the authors have adequately addressed your comments raised in a previous round of review and you feel that this manuscript is now acceptable for publication, you may indicate that here to bypass the “Comments to the Author” section, enter your conflict of interest statement in the “Confidential to Editor” section, and submit your "Accept" recommendation.

Reviewer #2: All comments have been addressed

2. Is the manuscript technically sound, and do the data support the conclusions?

Reviewer #2: Yes

3. Has the statistical analysis been performed appropriately and rigorously? 

Reviewer #2: Yes

4. Have the authors made all data underlying the findings in their manuscript fully available?

Reviewer #2: Yes

5. Is the manuscript presented in an intelligible fashion and written in standard English?

Reviewer #2: Yes

6. Review Comments to the Author

Reviewer #2: The comments made earlier in the review I submitted on the manuscript titled: "Drug-related problems in hypertension and gestational diabetes mellitus: a hospital cohort" have now been addressed by the authors in the resubmission and the manuscript can now be accepted for publication.

7. PLOS authors have the option to publish the peer review history of their article (what does this mean?). If published, this will include your full peer review and any attached files.

Reviewer #2: **Yes: **Modupe Olufunmilayo Ogunrombi

---

## [Editor Report · Acceptance letter]

30 Mar 2023

PONE-D-22-28909R1 

Drug-related problems in hypertension and gestational diabetes mellitus: a hospital cohort. 

Dear Dr. Martins:

I'm pleased to inform you that your manuscript has been deemed suitable for publication in PLOS ONE. Congratulations! Your manuscript is now with our production department. 

Kind regards, 

on behalf of

Dr. Nnabuike Chibuoke Ngene 

Academic Editor

PLOS ONE